# Multimodal detection of dopamine by sniffer cells expressing genetically encoded fluorescent sensors

Carmen Klein Herenbrink[1,9], Jonatan Fullerton Støier [1,9], William Dalseg Reith[1], Abeer Dagra[2], Miguel Alejandro Cuadrado Gregorek[1], Reto B. Cola [3], Tommaso Patriarchi [3,4], Yulong Li [5,6,7], Lin Tian [8], Ulrik Gether [1] & Freja Herborg [1✉]

Dopamine supports locomotor control and higher brain functions such as motivation and learning. Consistently, dopaminergic dysfunction is involved in a spectrum of neurological and neuropsychiatric diseases. Detailed data on dopamine dynamics is needed to understand how dopamine signals translate into cellular and behavioral responses, and to uncover pathological disturbances in dopamine-related diseases. Genetically encoded fluorescent dopamine sensors have recently enabled unprecedented monitoring of dopamine dynamics in vivo. However, these sensors' utility for in vitro and ex vivo assays remains unexplored. Here, we present a blueprint for making dopamine sniffer cells for multimodal dopamine detection. We generated sniffer cell lines with inducible expression of seven different dopamine sensors and perform a head-to-head comparison of sensor properties to guide users in sensor selection. In proof-of-principle experiments, we apply the sniffer cells to record endogenous dopamine release from cultured neurons and striatal slices, and for determining tissue dopamine content. Furthermore, we use the sniffer cells to measure dopamine uptake and release via the dopamine transporter as a radiotracer free, high-throughput alternative to electrochemical- and radiotracer-based assays. Importantly, the sniffer cell framework can readily be applied to the growing list of genetically encoded fluorescent neurotransmitter sensors.

[1] Molecular Neuropharmacology and Genetics Laboratory, Department of Neuroscience, Faculty of Health and Medical Sciences, University of Copenhagen, Copenhagen, Denmark. [2] College of Medicine, University of Florida, Gainesville, FL 32611, USA. [3] Institute of Pharmacology and Toxicology, University of Zurich, Zurich, Switzerland. [4] Neuroscience Center Zurich, University and ETH Zurich, Zurich, Switzerland. [5] State Key Laboratory of Membrane Biology, Peking University School of Life Sciences, 100871 Beijing, China. [6] PKU-IDG/McGovern Institute for Brain Research, 100871 Beijing, China. [7] Peking-Tsinghua Center for Life Sciences, 100871 Beijing, China. [8] Departments of Biochemistry and Molecular Medicine, School of Medicine, University of California, Davis, Davis, CA, USA. [9] These authors contributed equally: Carmen Klein Herenbrink, Jonatan Fullerton Støier. ✉email: frejahh@sund.ku.dk

D opamine (DA) serves as a neuromodulator in the brain where it is critically involved in locomotor control and higher brain functions such as motivation and reward-related learning. In line with these functions, decades of research have implicated disturbances in dopaminergic neurotransmission in both movement disorders and mental illnesses[1]. The dopaminergic circuit of the brain is also a major target for several therapeutics used in the treatment of these movement and mental disorders, and for drugs of abuse[2–5]. Still, we only have a limited understanding of the nature and progression of DA dysfunction in diseased states. In addition, the mechanisms through which DA exerts its short- and long-term effects on emotional states and behavior remain unclear. Sensitive methods to study DA neurotransmission in cell cultures, tissue preparations, and living organisms are necessary to gain mechanistic insights into DA signaling in health and disease states, and for the development of effective therapeutics that target dopaminergic circuits.

Important advances in the field of DA detection have recently been achieved with the development of G protein-coupled receptor (GPCR)-based sensors that directly couple the presence of DA with an increase in fluorescent. These sensors allow the interrogation of extracellular DA levels with unprecedented spatiotemporal resolution using optical measurements of fluorescent intensity[6,7]. Several studies have already demonstrated the powerful application of viral expression of such sensors for studying DA dynamics in live animals and in brain tissue circuits using e.g., fiber photometry and microscopy techniques[6–12]. Two families of GPCR-based DA sensors are currently available, the dLight and GRAB$_{DA}$ family, which are based on the DA D$_1$ receptor (D1R) and DA D$_2$ receptor (D2R), respectively. While both of these sensor families are based on the coupling of inert DA receptors to a conformational sensitive circularly permuted GFP molecule, they encompass distinct properties[6,7,13,14]. Overall, the sensors fulfill a number of attractive features such as high molecular specificity and affinities similar to that of endogenous DA receptors, large dynamic ranges, and a single-fluorophore protein design that allows for viral delivery and cell-specific expression[6,7,13,14]. However, the sensors have different intrinsic properties that should be carefully considered in the context of experimental conditions and the research question, but a direct side-by-side comparison of dopamine sensors is currently lacking to guide neuroscientists in choosing the most suitable sensor. In addition, the potential use of the growing list of GPCR-based sensors for the development of assays for in vitro and ex vivo DA recordings is largely unexplored.

Here, we present an easy-to-use, inexpensive, and scalable framework for applying GPCR-based DA sensors for multimodal in vitro and ex vivo measurements of DA. We establish seven different DA sensing (sniffer) cell lines with inducible expression of four dLight and three GRAB$_{DA}$ sensors and carry out a head-to-head comparison of sensor properties under identical experimental conditions. We perform proof-of-principle experiments showing how such sniffer cells can readily be applied to record the release of endogenous DA from cultured neurons and striatal slices and to determine total DA content in striatal tissue. Moreover, we demonstrate that the sniffer cells also enable measurements of DA transporter (DAT) activity, such as DA uptake and efflux, allowing for a radiotracer-free, high-throughput alternative to electrochemical- and radiotracer-based assays. Importantly, this framework for versatile usage of DA sniffer cells can easily be applied to other transmitter systems for which the palette of genetically encoded single-fluorescent protein sensors is continuously expanding. Because of the ease of use, low costs, and virus- and radioactivity-free properties, fluorescent sensor-expressing sniffer cells have great potential for becoming a general tool for studying transmitter levels in culture systems and tissue preparations.

## Results and discussion

**Development and characterization of DA sniffer cell lines.** With the innovative development of GPCR-based DA sensors[6,7], we wanted to expand the toolbox for DA detection in vitro and ex vivo with a virus and radiotracer-free method that allows for DA detection across multiple assay and sample formats using commonly available plate readers and fluorescent microscopes. To do this, we generated DA sensing sniffer cells by stable transfection of Flp-In T-REx 293 cells with DA sensors of either the dLight or the GRAB$_{DA}$ sensor family. Seven DA sensors were selected for this study: dLight1.1, dLight1.2, dLight1.3a, dLight1.3b, GRAB$_{DA1M}$, GRAB$_{DA1H}$, and GRAB$_{DA2M}$. Of note, as the Flp-In system was utilized for the generation of the sniffer cell lines, all sensors were inserted at the same specific genomic location ensuring homogenous levels of gene expression[15].

We first validated the sniffer cell lines using fluorescent microscopy to ensure tetracycline-induced expression of the sensors and confirm DA sensitivity. Indeed, upon treatment with tetracycline, all cell lines expressed their respective sensor, and all displayed an increase in fluorescent upon incubation with 10 µM DA (Fig. 1a). We then carried out a head-to-head comparison of the seven different DA sniffer cell lines to derive key sensor properties under identical experimental conditions. The most important sensor properties one needs to consider are the dynamic range, sensor sensitivity, and kinetic parameters, which need to be compatible with the expected DA concentrations and fluctuations of the model system.

To determine the dynamic range, we applied fluorescent microscopy to measure the change in fluorescent (F/F$_0$) following the application of 10 µM DA to the sniffer cells (Fig. 1a and Table 1). The greatest dynamic range was observed for dLight1.3b (F/F$_0$ = 6.61 ± 0.47) followed by the dLight1.3a (F/F$_0$ = 4.98 ± 0.24) and GRAB$_{DA2M}$ (F/F$_0$ = 4.77 ± 0.22) sensors, while dLight1.1 (F/F$_0$ = 2.29 ± 0.06) and GRAB$_{DA1M}$ (F/F$_0$ = 1.86 ± 0.07) sensors showed the lowest dynamic range. dLight1.2 and GRAB$_{DA1H}$ had a fluorescent change (F/F$_0$) of 3.16 ± 0.20 and 2.49 ± 0.04, respectively. The dynamic range for the D2R-derived sensors (GRAB$_{DA}$ family) was similar to the previous studies[7,16]. We also observed comparable, albeit slightly smaller, dynamic ranges in the sniffer cells expressing the D1R-derived sensors (dLight family) as compared to what was previously reported[6]. The small changes in dynamic ranges can likely be explained by a difference in background fluorescent (F$_0$) as a result of the differential experimental setup and/or differences in expression levels of the sensors. Importantly, however, the relative difference in F/F$_0$ between the four dLight sensors appears to be consistent between the studies, with for instance dLight1.3a showing a two-fold greater F/F$_0$ than dLight1.1.

Next, we characterized the sensors' DA sensitivity by exposing the sniffer cells to increasing DA concentrations. The fluorescent change was detected with both a fluorescent microscope and plate reader to determine the detection range (Figs. 1b, 2a and Supplementary Figs. 1, 2). As is evident from the concentration-response curves in Fig. 2a, the different sensors can detect DA at a wide range of concentrations. The sniffer cells expressing the D1R-derived sensors have a detection range of 40 nM to 17 µM, with the dLight1.1 sniffer cells being the most sensitive (Table 1). The D2R-derived sensors, on the other hand, were more sensitive to lower concentrations of DA, which is consistent with the D2R having a higher DA affinity than the D1R. The GRAB$_{DA1M}$ and GRAB$_{DA2M}$ sniffer cells had a detection range of 4 nM to 1.8 µM DA, whereas the GRAB$_{DA1H}$ sniffer cells were able to detect DA levels as low as 1 nM (Table 1). Importantly, we confirmed that the increases in fluorescent were mediated via the sensors, as their responses were blocked by selective DA receptor antagonists (Fig. 2c, d). Overall, the observed detection ranges were similar

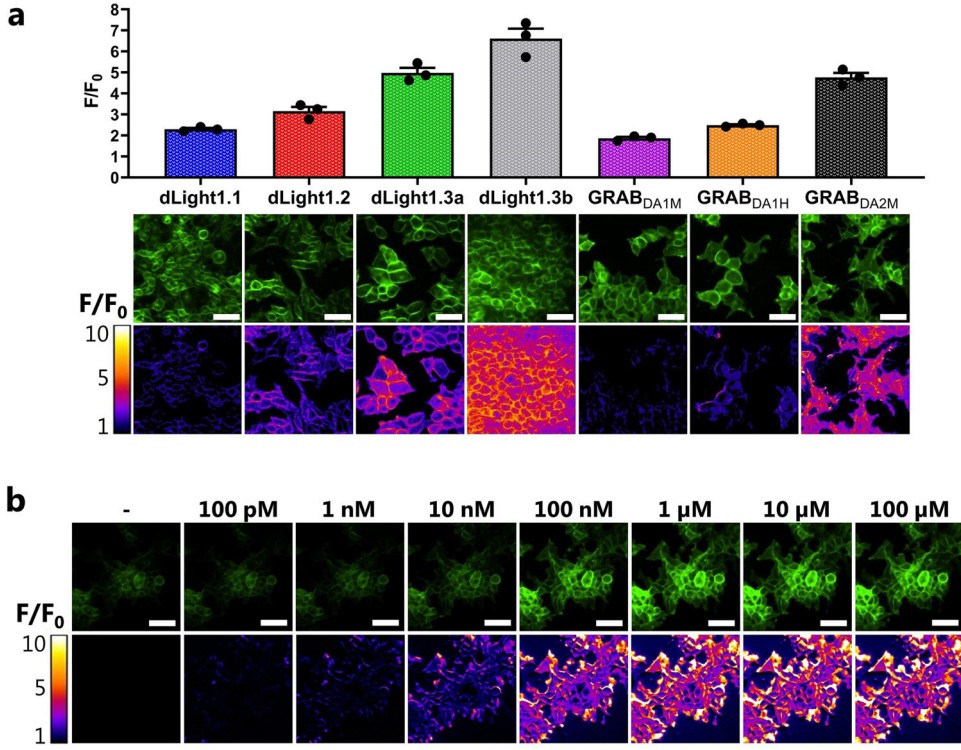

**Fig. 1 Characterization of the dynamic range of DA sniffer cell lines. a** Visualization and quantification of the fluorescent change in Flp-In T-REx-293 cells expressing one of seven different DA sensors. Cells were stimulated with 10 μM DA while imaged on an epifluorescent microscope ($N = 3$, mean ± SEM). Representative images of the raw fluorescent signal are shown along with corresponding pseudo-colored representations of the fold change above baseline. Scale bars are 50 μM. **b** Representative images of GRAB$_{DA2M}$ sniffer cells stimulated with increasing doses of DA show a dose-dependent increase in fluorescent ($N = 3$). Similar dose-response images of the remaining sniffer cell lines can be found in Supplementary Figs. 1, 2. Scale bars are 50 μM.

**Table 1 DA potency, detection range, dynamic range, and kinetic parameters of DA sensors.**

| Sensor | pEC$_{50}$ ± SEM (nM)[a] | Detection range 10–90% (nM)[a] | Dynamic range F/F$_0$ ± SEM[b] | Kinetic parameters $k_{on}$ (M$^{-1}$ min$^{-1}$)[a] | $k_{off}$ (min$^{-1}$)[a] |
|---|---|---|---|---|---|
| dLight1.1 | 6.45 ± 0.12 (350) | 40–3100 | 2.29 ± 0.06 | 1.84 ± 0.76 × 10$^8$ | 120 ± 17 |
| dLight1.2 | 5.92 ± 0.04 (1200) | 140–10,000 | 3.16 ± 0.20 | 7.50 ± 0.59 × 10$^7$ | 130 ± 17 |
| dLight1.3a | 5.87 ± 0.06 (1300) | 130–14,000 | 4.98 ± 0.24 | 6.42 ± 0.72 × 10$^7$ | 150 ± 4.8 |
| dLight1.3b | 5.67 ± 0.02 (2100) | 190–17,000 | 6.61 ± 0.47 | 7.17 ± 1.60 × 10$^7$ | 115 ± 10 |
| GRAB$_{DA1M}$ | 7.12 ± 0.04 (75) | 4.0–1400 | 1.86 ± 0.07 | 6.60 ± 1.66 × 10$^8$ | 80 ± 6.0 |
| GRAB$_{DA1H}$ | 8.34 ± 0.19 (4.6) | 0.78–27 | 2.49 ± 0.04 | ND | 9.2 ± 0.42 |
| GRAB$_{DA2M}$ | 6.90 ± 0.07 (130) | 8.6–1800 | 4.77 ± 0.22 | 3.14 ± 0.38 × 10$^8$ | 45 ± 4.3 |

*ND* Not determine due to ligand depletion.
[a]Determined with a fluorescent plate reader.
[b]Determined with a fluorescent microscope.

between plate reader and microscope experiments and in line with previous studies[6,7,13]. The dopamine sensors are not only capable of binding DA but also noradrenaline (NA)[6,7]. As detection of NA could confound data obtained from samples that also contains NA, we sought to determine the potency of this neurotransmitter for all seven sensors to determine the molecular selectivity of the various sensors, i.e., the extent of the selectivity of DA over NA (Fig. 2b and Table 2). Importantly, all sensors were preferentially activated by DA over NA. The fold selectivity (EC$_{50}$(NA/DA)) for DA over NE for the dLight family was 12 (dLight1.1), 13 (dLight1.2), 18 (dLight1.3a), and 16 (dLight1.3b). The GRAB$_{DA}$ family showed a wider range of DA selectivity, with a 21-fold (GRAB$_{DA1M}$), 8-fold (GRAB$_{DA1H}$), and 14-fold (GRAB$_{DA2M}$) selectivity for DA over NE (Table 2).

Activation and deactivation kinetics of a sensor are pivotal parameters to consider for experiments where detection of DA fluctuations at high temporal resolution is important. For instance, temporal resolution is critical to capture the rapid dynamics of DA release and clearance from dopaminergic neurons. To gain greater insight into the kinetics of the sensors utilized in this study, we determined the on ($k_{on}$) and off ($k_{off}$) activation rates by DA at the various sensors. To do so, we stimulated the sniffer cells with two relatively low concentrations of DA (to ensure that the kinetics are activation and not diffusion driven) and measured the change in fluorescent over time (Table 1 and Supplementary Fig. 3a–f). It should be noted that previous studies have already determined time (τ) or half-life (t$_{1/2}$) constants for some of the sensors utilized in our study. However, as the on-rate is dependent on the DA

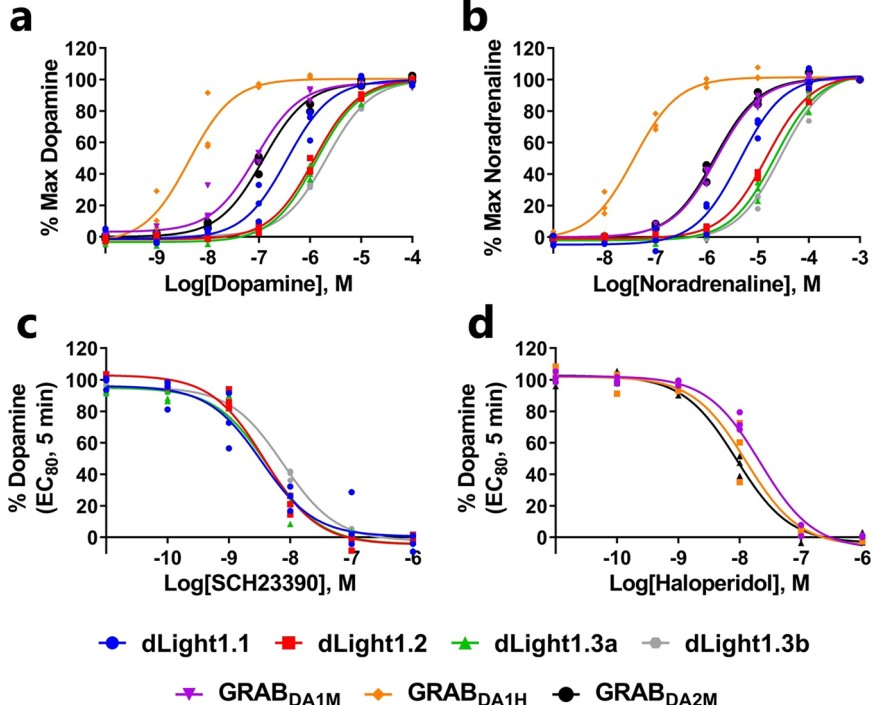

**Fig. 2 Characterization of the detection range of DA sniffer cell lines. a, b** DA (**a**) and NA (**b**) dose-response curves for the seven DA sniffer cell lines recorded after 5 min stimulation at 37 °C with indicated DA and NA concentrations, respectively, using a fluorescent plate reader. **c, d** The antagonists SCH23390 (**c**) and haloperidol (**d**) dose-dependently blocked the DA-induced increase in fluorescent in sniffer cells expressing the DA $D_1$ (dLight) and $D_2$ (GRAB) receptor-derived sensors, respectively, as detected by a fluorescent plate reader. Data were shown as fitted curves in scatter plots from three independent experiments.

**Table 2 NA potency and the selectivity of DA over NA of DA sensors.**

| Sensor | pEC$_{50}$ ± SEM (nM)$^a$ | Selectivity EC$_{50}$(NA/DA)$^a$ |
|---|---|---|
| dLight1.1 | 5.36 ± 0.07 (4400) | 12 |
| dLight1.2 | 4.80 ± 0.03 (16000) | 13 |
| dLight1.3a | 4.62 ± 0.05 (24000) | 18 |
| dLight1.3b | 4.47 ± 0.07 (34000) | 16 |
| GRAB$_{DA1M}$ | 5.80 ± 0.06 (1600) | 21 |
| GRAB$_{DA1H}$ | 7.43 ± 0.07 (37) | 8 |
| GRAB$_{DA2M}$ | 5.84 ± 0.05 (1400) | 12 |

$^a$Determined with a fluorescent plate reader.

concentration, such values cannot be compared between sensors. As shown in Table 1, we found that the D1R-based dLight sensors displayed slower on-rates ($k_{on}$) than the D2R-based GRAB$_{DA1M}$ and GRAB$_{DA2M}$ sensors. The off-rates ($k_{off}$), however, were faster for the dLight sensors than the GRAB$_{DA}$ sensors (Table 1). Through two-photon imaging, Patriarchi et al. (2018)[6] previously determined ex vivo that the dLight1.1 and dLight1.2 sensors display decay half-life constants of 100 and 90 ms in the dorsal striatum, respectively[6]. We derived off-rates ($k_{off}$) of 120 ± 17 and 130 ± 17 min$^{-1}$ for dLight1.1 and dLight1.2, which equal to decay half-life constants ($t_{1/2}$) of 340 and 324 ms ($t_{1/2}$ = ln2/$k_{off}$), respectively. The slower off-rates observed in our study may arise from the differential experimental conditions under which the parameters were obtained i.e., plate reader recordings of HEK293 cells activated by DA addition versus two-photon imaging of dopamine released upon electrical stimulation of brain slices. The off-rates of GRAB$_{DA1M}$ ($k_{off}$ = 80 ± 6.0 min$^{-1}$) and GRAB$_{DA2M}$ ($k_{off}$ = 45 ± 4.3 min$^{-1}$)

were similar to what has previously been reported (85 and 46 min$^{-1}$, respectively ($k = \tau^{-1}$))[7,16]. Unfortunately, we were not able to obtain accurate on- and off-rates for GRAB$_{DA1H}$ likely due to ligand depletion caused by the high sensitivity and expression of the sensor in combination with a small assay volume. To overcome ligand depletion, we determined the dissociation kinetics in the presence of a high concentration of the D2R antagonist haloperidol instead (Supplementary Fig. 3g). We obtained an off-rate of 9.2 ± 0.42 min$^{-1}$ for the GRAB$_{DA1H}$ sensor which was slower than what previously has been reported (24 min$^{-1}$ ($k = \tau^{-1}$))[7,16].

In summary, our head-to-head comparison of DA sensors on the detection range, dynamic range, and kinetic parameters should serve as a guideline for users to select the most appropriate DA sensor to use in their specific experiments. In assays with poor signal-to-noise ratio, the large dynamic range of dLight1.3a, dLight1.3b, and GRAB$_{DA2M}$ is necessary to accurately measure extracellular DA levels. On the other hand, high DA sensitivity is an attractive property for assays that require the detection of low DA concentrations. For example, if one needs to measure very low levels of extracellular DA (<4 nM), GRAB$_{DA1H}$ is one of the few sensors currently available that has the required DA sensitivity. However, the increased DA sensitivity is associated with a slower off-rate, which reduces the temporal detection accuracy due to the integration of temporally close release events. Thus, for in vivo and ex vivo experiments where it is paramount to capture the rapid firing events of DA neurons and where DA levels are sufficiently high, the fast off-rate kinetics of the dLight sensors are favored. Another important consideration related to the sensors' kinetic properties is their potential buffering effect, where DA availability to endogenous receptors is altered by the sensor expression. High expression of sensors, particularly with slow off-rates, may buffer a significant fraction of extracellular DA and blunt fast changes in dopamine levels, which could

influence cellular or behavioral responses. Collectively, the distinct properties of each DA sensor should be carefully considered and matched to the technical setup and biological conditions.

**DA sniffer cells: a tool for studying DAT pharmacology and function.** DAT plays an important role in DA homeostasis as it rapidly clears DA from the extracellular space to the cytoplasm for subsequent storage and release. DAT is also the primary target for both illicit substances (e.g., psychostimulants such as cocaine and methamphetamine) and therapeutic agents such as amphetamine (AMPH) and methylphenidate used for the treatment of ADHD[17]. We wanted to assess if the DA sniffer cells could be used as a tool to study DAT function. For this, we transiently transfected GRAB$_{DA2M}$ sniffer cells with human DAT (hDAT). We rationalized that hDAT-mediated DA uptake would produce a local decrease in extracellular DA concentration near the cell surface and thereby reduce activation of nearby GRAB$_{DA2M}$. Thus, exposure to dopamine should produce a smaller fluorescent change in hDAT-expressing GRAB$_{DA2M}$ sniffer cells than in GRAB$_{DA2M}$ sniffer that were mock-transfected with an empty vector. To test this, we plated GRAB$_{DA2M}$ sniffer cells that were transiently transfected with hDAT or empty vector as control and recorded the GRAB$_{DA2M}$ fluorescent in a plate reader during stimulation with 1 µM DA (Fig. 3a and Supplementary Fig. 4). To ensure that the measurements between hDAT-and empty vector-transfected cells were comparable (as the expression of hDAT could affect the sensor expression) the data were normalized to stimulation with a saturating concentration of DA (10 µM). As expected, we observed a markedly lower fluorescent signal in cells expressing hDAT than in the mock-transfected cells, reflecting hDAT-dependent DA uptake. To ensure that the reduced signal was truly mediated by hDAT and not an artifact of decreased surface expression levels of the sensor, the experiment was repeated in the absence and presence of the DAT blockers nomifensine and cocaine (Fig. 3b). Indeed, preincubation with either blocker dose-dependently reversed the decreased fluorescent change upon addition of DA with a pIC$_{50}$ of 6.09 ± 0.07 (820 nM) for cocaine and 6.77 ± 0.06 (170 nM) for nomifensine, which is comparable to earlier findings[18]. Collectively these data demonstrate that the sniffer cells can be used as a radiotracer-free alternative for measuring DAT-dependent DA uptake and for screening DAT blockers to derive indirect measures of apparent affinities.

While DAT normally functions through inward transport of DA, studies on the mechanisms of psychostimulants have revealed that DAT can mediate reverse transport of dopamine as well and that this efflux is essential for the action of AMPH[19,20]. Additionally, it has been shown that certain disease-associated mutations in DAT can cause anomalous constitutive DA efflux, which compromises the ability to accumulate DA[21–24]. We examined whether the sniffer cells could be used as an approach for detecting hDAT-mediated DA efflux. First, we determined if we could observe AMPH-induced DA efflux in hDAT-transfected GRAB$_{DA2M}$ sniffer cells in a high-throughput plate reader format. For this, sniffer cells were transfected with either hDAT or an empty vector and loaded with 300 nM DA for 15 min, after which extracellular DA was removed. We then added either vehicle or 10 µM nomifensine (10 min) before stimulating the cells with increasing concentrations of AMPH (10 nM–10 µM). As seen in Fig. 3, AMPH elicited a rapid dose-dependent increase in extracellular DA levels, measured as an increase in GRAB$_{DA2M}$ fluorescent in hDAT-transfected cells, which was blocked by preincubation with nomifensine (Fig. 3c, d). Moreover, from the dose-response

curves, we derived a pEC$_{50}$ value of 6.32 ± 0.09 (476 nM) for AMPH (Fig. 3d), which is comparable to what has previously been reported for hDAT-expressing heterologous cells[25,26]. It should be noted that the addition of nomifensine to hDAT-expressing GRAB$_{DA2M}$ sniffer cells also increased the extracellular DA concentration, although markedly less than AMPH (Fig. 3c). This increase presumably reflects DAT-independent DA leakage from the cells that can no longer be transported back into the cells by hDAT when blocked by nomifensine[27]. Importantly, nomifensine and AMPH had no effect in GRAB$_{DA2M}$ sniffer cells co-transfected with an empty vector, confirming that the observed effects are mediated through hDAT and not due to a direct interaction between the drugs and the sensor.

We then determined whether the sniffer cells could be used to validate an aberrant molecular phenotype, which has been described for an autism-associated de novo variant, hDAT-T356M. The hDAT-T356M variant imposes conformational changes to DAT that causes a leak of DA through the transporter (an anomalous DA efflux), which can be inhibited by DAT blockers[21,28]. To study this phenomenon with the sniffer cells, GRAB$_{DA2M}$ sniffer cells were transiently transfected with either WT hDAT or hDAT-T356M, and loaded with a high DA concentration (10 µM) for 15 min. The cells were then washed and equilibrated for 20 min after which they were treated with nomifensine (10 µM) to block both reuptake and anomalous DA efflux. As expected, nomifensine caused an increase in extracellular DA in WT DAT transfected cells as it blocks the reuptake of DAT-independent DA leakage. In contrast, stimulation of T356M-transfected sniffer cells with nomifensine produced a remarkable decrease in extracellular DA, consistent with blockage of constitutive DA efflux via hDAT-T356M (Fig. 3e, f). The phenomenon of anomalous DA efflux has been proposed to be a common mechanism through which missense mutations in DAT may exert disturbances in DA neurotransmission that are of pathophysiological relevance[24]. So far, investigations of DAT-mediated DA efflux have relied on amperometric recording and superfusion assays with $^3$H-MPP$^+$[21–23,29]. Our data show that sniffer cells can be applied as an alternative, less labor-intensive strategy for identifying disturbances in DAT efflux properties and for studying the molecular and cellular consequences.

**Detection of DA release from cultured dopaminergic neurons and striatal slices, and quantification of DA content in brain tissue.** Having established potential applications of the sniffer cells in heterologous cell assays, we next sought to explore if the sniffer cells could be applied to visualize the release of endogenous DA from cultured dopaminergic neurons, and for ex vivo measurements of DA release from striatal slices and quantifications of DA tissue content in striatal homogenates.

To visualize DA release from cultured rat dopaminergic neurons, GRAB$_{DA1H}$ sniffer cells were seeded on top of cultured neurons 24 h prior to experiments (Fig. 4a). Wide-field fluorescent imaging was then conducted in aCSF (artificial cerebrospinal fluid) under constant slow perfusion. We stimulated the neurons with electrical field stimulation (Fig. 4b) to induce DA release and recorded the change in fluorescent of the GRAB$_{DA1H}$ sniffer cells. Upon stimulation, an instantaneous increase in extracellular DA levels was detected. Termination of the electrical field stimulation reverted the fluorescent change back to baseline (Fig. 4b). The decrease in fluorescent signal is presumably a combination of the speed of the perfusion system, the reuptake kinetics of DA back into the neurons, and the off-rate of the GRAB$_{DA1H}$ sensor.

We also tested the applicability of using the sensor cells to record from scarcely seeded mouse dopaminergic neurons that

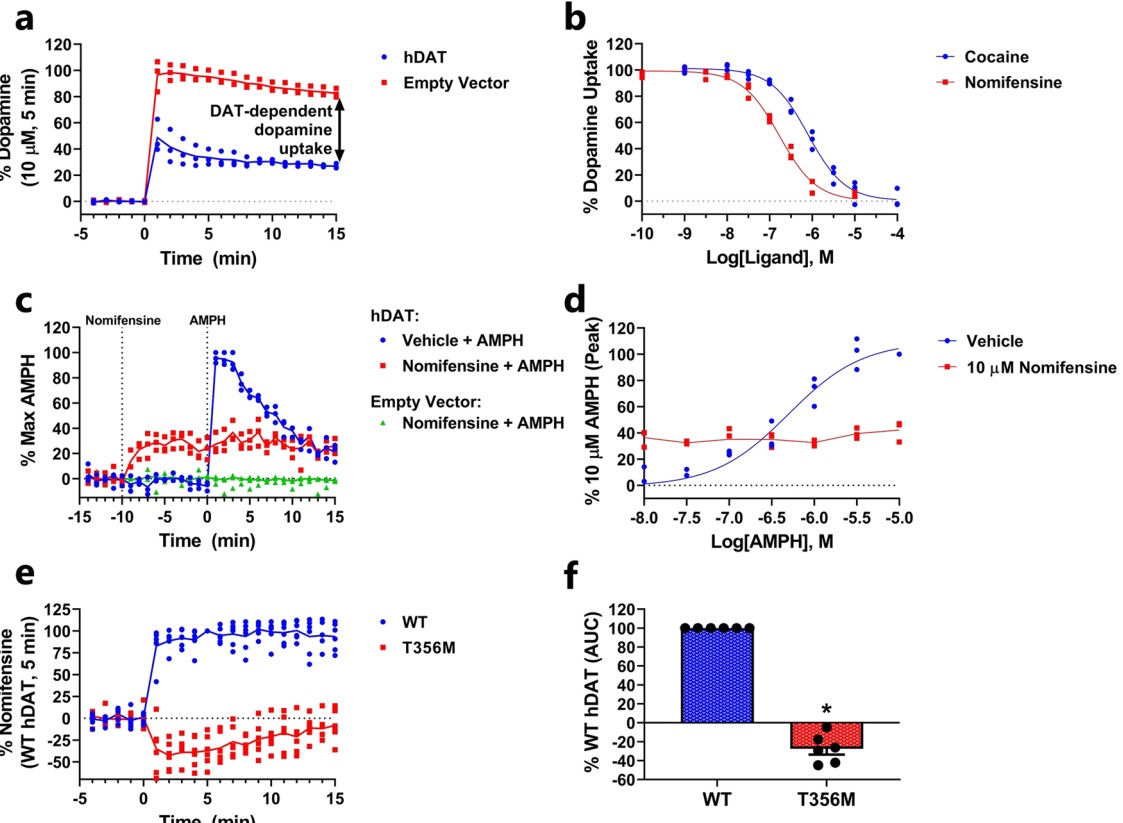

**Fig. 3 Detection of DAT-mediated DA uptake and efflux using sniffer cells. a** Measurement of DA uptake using DA sniffer cells. GRAB_DA2M sniffer cells, transfected with hDAT, or an empty pcDNA3.1 expression vector, were stimulated with 1 μM DA for 15 min. The decreased fluorescent upon addition of DA to cells transfected with hDAT versus an empty vector is indicative of DAT-mediated DA uptake into the cells. To ensure that the measurements between hDAT and empty vector-transfected cells were comparable (as the expression of hDAT could affect the sensor expression) the data were normalized to a 5 min stimulation with a saturating concentration of DA (10 μM). The change in fluorescent upon DA addition is shown with a higher temporal resolution in Supplementary Fig. 4. **b** Preincubation of hDAT-transfected GRAB_DA2M sniffer cells with cocaine or nomifensine dose-dependently decreased the hDAT-mediated DA uptake. **c, d** Measurement of AMPH-induced DA efflux via DAT using DA sniffer cells. GRAB_DA2M sniffer cells transfected with hDAT or an empty pcDNA3.1 expression vector were loaded with 300 nM DA and washed subsequently. After reaching equilibrium, the baseline was recorded and the sniffer cells were incubated with 10 μM nomifensine or vehicle for 10 min followed by a 15 min stimulation with AMPH (**c**, 10 μM). AMPH caused a dose-dependent increase in fluorescent in hDAT-transfected cells, which was absent when the cells were preincubated with 10 μM nomifensine (**d**). **e, f** Recordings of anomalous DA efflux by the disease-associated DAT-T356M mutant. GRAB_DA2M sniffer cells were transfected with WT hDAT or hDAT-T356M. After loading the cells with 10 μM DA and subsequently washing away extracellular DA, the cells were stimulated with 10 μM nomifensine which blocks the constitutive anomalous DA efflux via hDAT-T356M. **f** Shows mean ± SEM area under the curve (AUC) for DAT-T356M relative to WT ($N = 6$, *$P < 0.05$ (<0.0001), one-sample $t$-test). The data were expressed as normalized Δ(F/F$_O$). Curves are presented as connecting lines of the mean (**a, c, e, d** (nomifensine)) or fitted curves (**b, d** (vehicle)) in scatter plots from three independent experiments conducted on a fluorescent plate reader.

expressed tdTomato to allow the identification of individual dopaminergic neurons (Fig. 4c and Supplementary Video 1). Upon depolarization with KCl, we observed an immediate but smaller increase in extracellular DA levels. While the change was detectable and significant, future studies aimed at detecting DA release from scarcely seeded neurons may benefit from enhancing the expression of GRAB_DA1H to gain higher sensitivity.

Next, we tested if we could employ the sniffer cells to detect DA release from acute striatal slices. Striatal slices were submerged in aCSF and incubated for 5 min with either KCl or AMPH to induce DA release, or with vehicle. The buffer was then collected and added to GRAB_DA2M sniffer cells already seeded into 96- well plates and the change in fluorescent was detected with a fluorescent plate reader (Fig. 4d). As expected, a greater increase in fluorescent was observed upon the addition of media collected from KCl- and AMPH-treated slices than from vehicle-treated samples (Fig. 4d).

Finally, we wanted to evaluate whether the sniffer cells could be used to determine DA content in mouse striatal tissue as an alternative approach to high-performance liquid chromatography (HPLC) analysis. To do so, we prepared tissue homogenates from the striatum in the absence of chemicals that could potentially affect the sniffer cells. As a negative control, we also prepared homogenates from the cerebellum. In order to determine the DA concentration, the homogenates were added to GRAB_DA2M sniffer cells together with a DA standard curve (for interpolation), and the change in fluorescent was detected with a fluorescent plate reader. The average DA level detected in the striatal samples was 33.5 ± 7.6 ng/mg protein (mean ± S.E.M.; eight mice) while the DA level in the cerebellum was below the detection limit of the GRAB_DA2M sniffer cells (Fig. 4e). The amount of DA detected in the striatal samples was within range of what has previously been reported by studies that utilized HPLC[30–33]. Thus, the sniffer cells provide an alternative approach for determining total

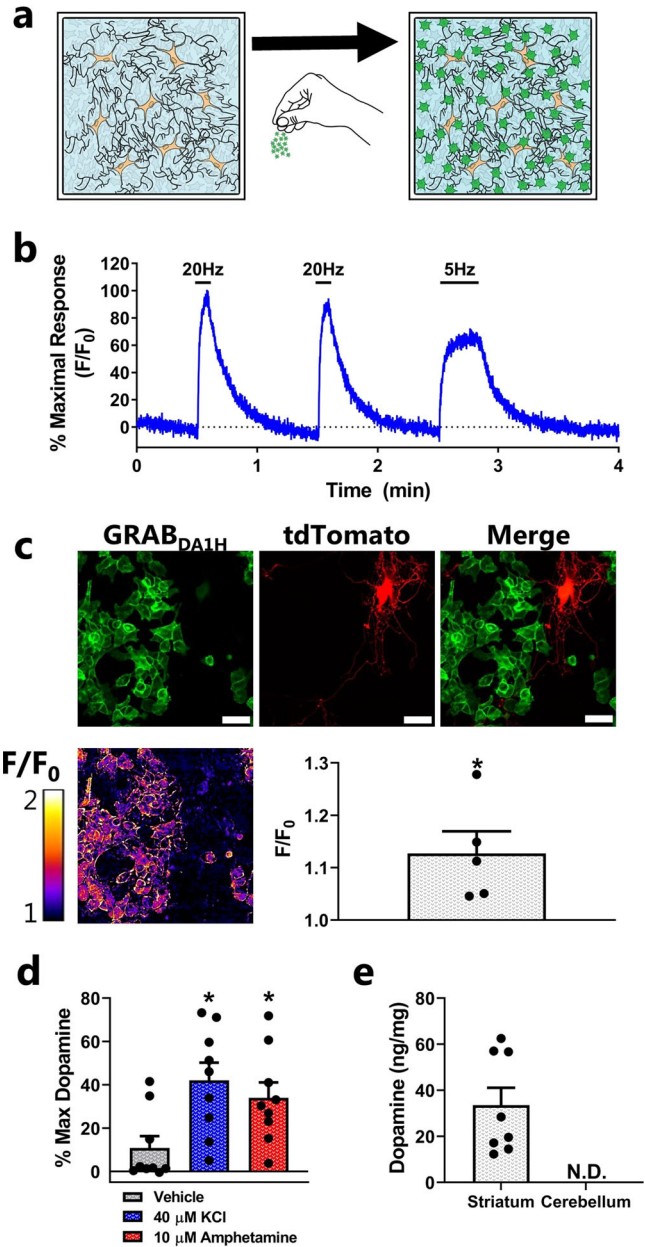

**Fig. 4 Detection of DA release from cultured dopaminergic neurons and striatal slices, and quantification of DA content in brain tissue. a** In order to detect DA release from cultured dopaminergic neurons, GRAB$_{DA1H}$ sniffer cells were plated on top of the neurons 24 h prior to experiments. **b** DA release from rat dopaminergic neurons evoked by electric field stimulation of 100 depolarizing monopolar pulses at a frequency of 20 or 5 Hz. DA release was detected as an increase in fluorescent from co-cultured GRAB$_{DA1H}$ sniffer cells. The trace is representative of three independent experiments, and the data were expressed as F/F$_0$ in % of the maximal response induced by the electrical field stimulation. **c** DA release from scarcely seeded mouse dopaminergic neurons expressing tdTomato for identification of individual neurons. Depolarization was induced by stimulation with 90 mM KCl. The representative images of the GRAB$_{DA1H}$ sniffer cells (green) and the tdTomato-expressing dopaminergic neuron (red) are shown together with corresponding representative pseudo-colored representations of the fold change above baseline in response to 90 mM KCl. The fluorescent change detected in five recordings from three independent neuronal cultures was quantified (mean ± SEM). *$P < 0.05$ (=0.017), one-sample $t$-test. **d** Mouse striatal slices, incubated for 5 min (37 °C) with a vehicle, 40 μM KCl, or 10 μM AMPH in a small volume of aCSF. DA release was determined by transferring the aCSF incubation buffer to a 96-well plate seeded with GRAB$_{DA2M}$ sniffer cells. DA content in the vehicle-, KCl-, and AMPH-treated samples was detected on a fluorescent plate reader, and the values are expressed as Δ(F/F$_0$) normalized to the maximal Δ(F/F$_0$) elicited by saturating [DA]. The values are the mean ± SEM from striatal slices of nine mice. *$P < 0.05$ ($P = 0.0086$ for KCl and $P = 0.0016$ for AMPH), one-way ANOVA with Dunnett post hoc test. **e** Total DA content was determined in the striatum and cerebellum of mice utilizing the GRAB$_{DA2M}$ sniffer cells. The tissue was homogenized, sonicated, and freeze/thawed in a hypotonic buffer to ensure that all DA was released from the tissue. Following centrifugation, the DA and protein concentrations were determined in the supernatant with a fluorescent plate reader allowing conversion to ng DA per mg protein. The DA levels in the cerebellum were too low to be detectable by the GRAB$_{DA2M}$ sniffer cells. The values are the mean ± SEM from the tissue of eight mice.

tissue DA content. It should, however, be noted that even though the GRAB$_{DA2M}$ sensor has a 12-fold selectivity for DA over NA, we cannot exclude that the striatal tissue lysates contain detectable amounts of NA and likewise that the treatment of acute slices with either KCl or AMPH may have also induced release of detectable amounts of NA besides DA.

Taken together, the data presented show that sniffer cells can be used as an alternative method to visualize and measure DA release from dopaminergic neurons and striatal slices (e.g., to study the effect of drugs or genetic manipulations on DA release from dopaminergic release sites). Furthermore, the cells can also be utilized as a readily approachable way to determine total tissue DA content.

## Conclusions

Our study presents a framework for use of cells expressing genetically encoded fluorescent sensors as a virus- and radio-tracer-free, inexpensive, and scalable approach for multimodal

in vitro and ex vivo measurements of neurotransmitter levels. The sniffer cell strategy presented here should be a generalizable and easily applicable framework for other genetically encoded single-protein fluorescent sensors.

## Methods

**Cloning**. GRAB$_{DA1H}$, GRAB$_{DA1M}$, and GRAB$_{DA2M}$ were subcloned into the pcDNA5/FRT/TO vector (Invitrogen) by Sequence and Ligation Independent Cloning (SLIC)[34]. The vector was linearized by cutting with EcoRV restriction enzyme, and inserts with the GRAB$_{DA}$ sensors were generated by PCR with SLIC Fwd (5′-TGGAATTCTGCAGATATGGAGACAGACACACTC-3′) and Rev (5′-GCCACTGTGCTGGATTCAGCAGTGGAGGATCTT-3′) primers. dLight1.1, dLight1.2, and dLight1.3b were all subcloned from their parent vector into the pcDNA5/FRT/TO vector using the HindIII and NotI restriction sites present in all three vectors. To generate dLight1.3a, we performed site-directed mutagenesis on dLight1.1 in pcDNA5/FRT/TO vector using the overlapping Fwd (5′-ACAG-GATTGCTCAGAAACAGCTGAGCTCACTCATT-3′) and Rev (5′-AATGAGT-GAGCTCAGCTGTTTCTGAGCAATCCTGT-3′) primers carrying the desired insert mutation (p.K247_L248insQ)[35]. All constructs were subsequently sequence-verified to confirm correct insertion.

**Generation of sniffer cell lines**. Flp-In T-REx 293 cells were grown in DMEM supplemented with 10% (v/v) fetal bovine serum (FBS; Invitrogen), 100 μg/mL zeocin (ThermoFisher Scientific), and 15 μg/mL blasticidin (ThermoFisher Scientific). To generate sniffer cell lines, the parental cells were grown in T150 flasks (Corning) until 70% confluency. The media was then changed to DMEM supplemented with 10% (v/v) FBS, and the cells were transfected with 0.6 μg DA sensor in a pcDNA5/FRT/TO vector and 5.4 μg pOG44 with 18 μL Lipofectamine 2000 (Invitrogen) according to the manufacturer's protocol. Forty-eight hours after transfection, the cells were split 1:3 into a new T150 flask (Corning). After adherence of the cells, the media was changed to DMEM supplemented with 10%

(v/v) FBS, 200 µg/mL Hygromycin B (Sigma), and 15 µg/mL blasticidin. The media was changed twice a week until colonies that stably express the sensor were obtained. The expression of the sensors was induced 24–48 h prior to experiments with 1 µg/mL tetracycline (Sigma).

**Culturing of midbrain dopaminergic neurons**. Cultures of midbrain dopaminergic neurons on top of cortical astrocytes were made from P1-P2 Wistar rats or DAT-IRES-Cre mice (Jackson Laboratory)[36]. Briefly, tissue was dissected from the ventral midbrain and digested in a papain solution oxygenated with a carbogen (95% $O_2$ + 5% $CO_2$) at 37 °C for 30 min. The digested tissue was brought to a single cell suspension by trituration through pipette tips of increasingly smaller sizes and centrifuged at 500 g for 10 min. The neurons were resuspended in a prewarmed neuron medium (Neurobasal A (10888022, Gibco) with 1% GlutaMAX (35050061, Gibco), 2% B-27 plus (A3582801, Gibco), 200 µM ascorbic acid, 500 µM kynurenic acid, and 0.1% Pen-Strep solution (P0781, Sigma)).

The cells were plated in neuron medium in six-well plates on a monolayer of glia cells grown on poly-D-lysine coated coverslips (∅ = 25 mm). Two hours after plating neurons, rat glial cell-derived neurotrophic factor (SRP3239, Sigma) was added for a final concentration of 10 ng/mL. The cultures were used for experiments 14–21 days after the neurons were plated out. The neurons obtained from DAT-IRES-Cre mice were transduced 5 days post dissection with pAAV-FLEX-tdTomato (Addgene).

**Imaging experiments and analysis**. Most imaging was performed using an ECLIPSE Ti-E epifluorescent/TIRF microscope (NIKON, Japan) with a 488 nm laser (coherent, California, USA) and an S Plan Fluor ELWD 20X/0.45 ADM microscope objective (NIKON, Japan). A 525/40 nm bandpass filter was used for the emission light, which was then recorded using an iXon3 897 Electron Multiplying CCD camera (Andor, United Kingdom). However, the imaging of the neurons obtained from DAT-IRES-Cre mice was conducted on a Nikon Eclipse FN1 upright microscope (Nikon, Japan).

For dose-response and max-response images of the individual DA sniffer cell lines, we seeded the cells out in eight-well Lab-Tek™ II Chambered Coverglass (Nunc) at a density of 25,000 cells per well and added tetracycline. The following day, the cells were washed twice with PBS and recorded with stepwise increasing [DA] for 5 min at each condition at a frame rate of 0.2 Hz.

To record DA from neuron cultures, $GRAB_{DA1H}$ sniffer cells were seeded out at a density of 200,000 cells per well on top of primary DAergic neuron cultures accompanied with tetracycline 1–2 days before the experiment. On the microscope, the neurons were mounted in an RC-21BRFS Field Stimulation Chamber (Warner Instruments, USA) and the neurons were continuously perfused with aCSF (in mM: NaCl, 120; KCl, 5; glucose, 30; $MgCl_2$, 2; $CaCl_2$, 2; HEPES, 25; pH 7.40) at 37 °C. A Master-8 Pulse Generator (A.M.P.I., Israel) and an ISO-Flex Stimulus Isolator (A.M.P.I., Israel) were used to evoke DA release by passing 1 ms monopolar current pulses through the stimulation chamber electrodes to yield an electric field strength of ~40 V/cm. Images were recorded at a frame rate of 12.5 Hz.

All microscopy images were analyzed using the ImageJ-based Fiji software[37]. For calculating max-responses, we generated an averaged image of five consecutive images recorded for both conditions (before and one after adding 10 µM DA). Then, the pixel intensities over the same cellular cross-sections for each condition was quantified and single peak values (arising at cell membrane sections) were selected to calculate $F/F_0$. For each N, we pseudo-randomly selected a population of cells with a low and a high baseline fluorescent and included two intensity peaks for each population. Thus, each N represents the mean $F/F_0$ at four intensity peaks. The images of sniffer cells on neuron cultures were quantified by drawing a region of interest around a population of sniffer cells and measuring its intensity as a function of time using the build-in "Measure Stack" ImageJ macro.

**Fluorescent plate reader experiments and analysis**. Sniffer cells were plated at a density of 30,000 cells/well into poly-L-ornithine-coated white CulturPlate-96 plates (PerkinElmer) and induced with tetracycline 48 h prior to experiments. In experiments with hDAT-transfected sniffer cells, the cells were simultaneously transfected with 50 ng hDAT and 150 nL Lipofectamine 2000 (Invitrogen) per well according to the manufacturer's protocol. Fluorescent was measured (485/520) with a POLARstar OMEGA plate reader (Biotek), and all experiments were conducted at 37 °C.

For characterization of the sniffer cell lines, the cells were washed with aCSF and incubated in aCSF for 15 min at 37 °C in the absence or presence of the DA receptor antagonists SCH23390 or haloperidol. Upon measuring the baseline fluorescent intensity, the cells were incubated with vehicle or increasing DA concentrations, and the fluorescent intensity was measured after 5 min. To detect the rapid increase or decrease of fluorescent intensity upon the addition of DA or antagonist, the drugs were injected into the well by the POLARstar OMEGA plate reader (Biotek).

To detect uptake of DA via hDAT into the sniffer cells, hDAT- or pcDNA3.1-transfected $GRAB_{DA2M}$ cells were incubated for 15 min at 37 °C with aCSF with or without cocaine or nomifensine. Afterward, the cells were stimulated with 1 µM DA and the fluorescent intensity was measured for 15 min. The cells were also stimulated with 10 µM DA to use for data normalization.

AMPH-induced hDAT-mediated DA efflux experiments were performed by loading hDAT-transfected $GRAB_{DA2M}$ cells with 300 nM DA for 15 min at 37 °C.

The cells were then washed three times with aCSF for 3 min each. After a 20 min incubation at 37 °C, the cells were stimulated with vehicle or nomifensine for 10 min followed by the addition of AMPH. The fluorescent intensity was measured every minute throughout the experiment.

Experiments that allow the detection of constitutive hDAT-mediated DA efflux were performed in a similar manner as the AMPH-induced DA efflux experiments. However, the cells were loaded with 10 µM DA rather than 300 nM DA, and the cells were only stimulated with nomifensine and not with AMPH.

All experiments were performed at least three times with duplicate or triplicate determinations.

To analyze the data, the fluorescent intensity was divided by the mean baseline fluorescent intensity to obtain $F/F_0$. Subsequently, the average $F/F_0$ of control wells that solely received vehicle was subtracted to gain $\Delta(F/F_0)$. These values were then normalized as indicated. All data were analyzed using GraphPad Prism 9. The association and dissociation kinetics were determined utilizing the Graphpad function "Association kinetics (two ligand concentrations)".

**Detection of DA release from acute mouse striatal slices**. Acute brain slices were acquired from adult C57BL/6 male or female mice (16 ± 4 weeks). The animals were anesthetized with isoflurane and the brains were quickly harvested into ice-cold aCSF. Coronal striatal brain slices of 300 µm thickness were prepared on a LeicaVT1200 vibrating blade microtome. Slices were then transferred to oxygenated aCSF at room temperature and allowed to recover for at least 1 h before the experiment. Subsequently, the slices were transferred to 2 mL Eppendorf tubes containing 500 uL prewarmed (37 °C) aCSF in the presence or absence of 40 mM KCl or 10 µM AMPH. After 5 min incubation at 37 °C, the aCSF was collected to determine whether DA release had occurred.

To detect whether stimulation with KCl or AMPH-induced DA release from acute striatal slices, $GRAB_{DA2M}$ sniffer cells were utilized. The cells were washed with 200 µL aCSF and incubated for 15 min at 37 °C with 100 µL aCSF. After a baseline read, 100 µL aCSF collected from the slices was added to the cells, and the response was measured after 5 min on the POLARstar OMEGA plate reader. To normalize the data, cells were also stimulated with aCSF and 10 mM DA.

**Determination of DA content in mouse striatum and cerebellum**. Tissue samples from the striatum and cerebellum were obtained from adult C57BL/6 male or female mice (16 ± 4 weeks). Striatal tissue samples were isolated from coronal slices using a brain matrix and a puncher. Tissue samples from cerebellar slices were used as a negative control. The tissue was collected in hypotonic buffer (25 mM HEPES, pH 7.40 with KOH) containing 1 mM glutathione (Sigma) to prevent oxidation of DA[38]. To ensure extraction of all DA, the sample was homogenized using a syringe with a 27 G needle, sonicated, and freeze/thawed five times by alternating between 37 and −80 °C. The sample was then centrifuged for 30 min at 4 °C at 16 × g, and the supernatant was collected for DA and protein determination.

The protein concentration was determined with a standard BCA kit (Pierce). The DA concentration was determined utilizing $GRAB_{DA2M}$ sniffer cells plated in a white CulturPlate-96 plate (PerkinElmer). The cells were washed with 200 µL aCSF, and then incubated for 15 min at 37 °C in 100 µL aCSF. After a baseline read, 100 µL supernatant was added to the cells, and the response was measured after 5 min on the POLARstar OMEGA plate reader. To determine the DA concentration, additional wells with cells were also incubated with a range of DA concentrations prepared in the hypotonic buffer to allow interpolation.

**Statistics and reproducibility**. GraphPad Prism 8.0 (GraphPad Software, San Diego, CA) software was used for statistical analysis and data fitting. Data represent at least three independent experiments. Data were presented as mean ± SEM or as scatter plots with fitted curves or a connecting line of the mean overlaid. Two-sided one-way ANOVA with Dunnett post hoc test was used to test for group differences. Two-sided one-sample $t$-tests were applied to analyze data normalized to WT or baseline within each experiment. Differences were considered significant for *$P < 0.05$. Figure legends include information about statistics used, the number of independent experiments (biological replicates, $n$), and note exact $P$ values in parenthesis.

**Study approval**. Experimental procedures on adult C57BL/6 male or female mice (16 ± 4 weeks) adhered to the European guidelines for the care and use of laboratory animals, EU directive 2010/63/EU and were approved by the Danish Animal Experimentation Inspectorate.

**Reporting summary**. Further information on research design is available in the Nature Research Reporting Summary linked to this article.

## Data availability
Source data for graphs are available in Supplementary Data 1. The remaining information or data are available from the corresponding author upon reasonable request.

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

## Acknowledgements

We thank Anette Dencker Kaas for her excellent technical assistance. The work was supported by the Independent Research Fund Denmark—Medical Sciences (C.K.H.: 6110-00292B, U.G.: 7016-00325B, and U.G.: 9039-00437B). The Lundbeck Foundation (C.K.H.: R322-2019-1502, F.H.: R303-2018-3540, and U.G.: R276-2018-792). The European Research Council (ERC) under the European Union's Horizon 2020 research and innovation program (T.P.: 891959).

## Author contributions

C.K.H., J.F.S., and F.H. were responsible for the overall experimental design. M.A.C.G., C.K.H., J.F.S., T.P., and R.B.C. generated sniffer cells. C.K.H., J.F.S., W.D.R., A.D., and F.H. conducted the experiments. C.K.H. and F.H. wrote the manuscript. C.K.H., U.G., and F.H. provided funding. Y.L and L.T. developed and provided the sensors. All authors have contributed to the editing and review of the final manuscript.

## Competing interests

T.P., L.T., and Y.L. have filed patent applications on the sensor technology utilized in the study of which the value may be affected by this publication. The remaining authors declare no competing interests.
