## [Peer Review File · Communications Biology]

Reviewers' comments:

Reviewer #1 (Remarks to the Author):

COMMENTS FOR AUTHORS

Herenbrink et al. generated sniffer cell lines expressing six different dopamine sensors and compared them to guide users in sensor selection. They showed that the sniffer cells can record endogenous dopamine release from cultured neurons and striatal slices as well as tissue dopamine content. They apply the sniffer cells to measure dopamine uptake and release via the dopamine transporter.

I appreciate the authors' efforts to generate and validate the sniffer cells. However, the dopamine sensors are not new and the current data with the sniffer cells stably expressing the sensors are well expected. I would strongly recommend including new scientific information or new screening results of DAT blockers for the publication of this journal.

In Figure 3B, the F/F₀ response of the sniffer cells after 90 mM KCl is only 1.1. This is very strong stimulation and I am wondering the reasons.

Editing comments

- Throughout the manuscript, a space is required between numbers and units. For example, page 4: 10uM need to be 10 uM, page 6: figure legend, page 7: 4.3min⁻¹ to be 4.3 min⁻¹, etc
- Page 4: head-to head
- page 16: Hz to be Hz.

Reviewer #2 (Remarks to the Author):

In this paper, the authors established Sniffer cells expressing the previously published genetically encoded Dopamine sensors and demonstrated their Dopamine responses. They established Sniffer cells expressing six different green DA sensors and carefully quantified the response to DA. They have also successfully utilized Sniffer cells to visualize DA uptake and release by DAT. Finally, they show the release of DA from in vitro cultured neurons and from ex vivo cultured striatum using Sniffer cells.

I enjoyed reading this manuscript. Overall, the work is, for the most part, sound, and experiments are well-designed. I appreciate the evaluation of DA sensors under the same experimental formats, providing much-needed information to experimentalists. I have some minor suggestions for clarifying some concerns in the manuscript.

1.

All reported DA sensors had been shown to have cross-reactivity to noradrenaline (NA). In particular, the series of GRAB_DA, which are based on DRD2, exhibit only a little difference in sensitivity between DA and NA. Therefore, the data obtained in Figure 3 must also take into account not only DA but NA release from the striatum. If possible, it would be desirable to obtain a dose-response curve of six DA sensors for NA under the same conditions. If not, differences in sensitivity to NA should be mentioned in the revised manuscript.

2.

The authors should describe whether the Sniffer cells used in this study were single-cell cloned or a bulk population.

3.

In Figure 2, it is intriguing findings that the response of the DA reporters is reduced in Sniffer cells expressing DAT and that DA is released through DAT. However, it seems unlikely that only tens of

thousands of cells could rapidly uptake such a large amount of DA (probably 100 uL/well of 10 uM DA solution) to reduce the concentration of DA (Figure 2A). The authors should verify whether or not there is an artifact. For example, does the %Dopamine value just after DA stimulation approach 100% in DAT-expressing cells, if data are acquired at a higher sampling rate? If the conditioned medium 15 minutes after DA stimulation in DAT-expressing cells is collected and added to Sniffer cells that do not express DAT, will their response be about 30% of the %Dopamine value? If DAT-expressing and non-DAT-expressing cells are co-cultured and imaged under a microscope, is it possible to obtain the same level of response for DA sensors?

4.

Figure legend of Fig 2A says 1 uM DA, but the Y-axis of Fig 2A says 10 uM, 5 min. Which is correct and what is 5 min?

Response to reviewers

We appreciate the insightful and constructive comments from the reviewers and the overall positive sentiment. However, the reviewers raised some important issues, which we have addressed in the revised manuscript as described in the point-by-point response below.

We present the original reviewer comments in bold and have included a detailed description of our changes to the manuscript. These modifications are indicated by the lines at the left border of the text and non-grammatical changes are highlighted in yellow.

Reviewer 1:

“I appreciate the authors’ efforts to generate and validate the sniffer cells. However, the dopamine sensors are not new and the current data with the sniffer cells stably expressing the sensors are well expected. I would strongly recommend including new scientific information or new screening results of DAT blockers for the publication of this journal.”

We are pleased that the reviewer appreciates the sniffer cell concept. While we understand and appreciate the reviewer’s comments, we do not agree with the view that the study is not novel because the sensors “are not new”. The sensors used in our study were published in 2018 and, so far, they have not been used as demonstrated in our manuscript. We also disagree that our study does not carry its own weight because the method could have been expected to work. Often, even simple scientific experiments do not turn out as expected and, thus, experiments as those described in the manuscript are required to show the feasibility and broad applicability of the sniffer cell approach. Indeed, no one has yet articulated or proven that such a framework actually works. We would even argue that the simplicity of the sniffer cell framework is its major strength as it can easily be applied to a range of sensors and in numerous experimental designs. We believe accordingly that our data provide researchers with a novel tool that will greatly aid the identification of novel DAT blockers, but we find that such screening efforts are beyond the scope of this article. Instead, we chose to expand the study by including a characterization of sniffer cells expressing the dLight1.3b sensor. This sensor has gained wide use for in vivo dopamine investigations, and we therefore believe that it adds significant value for experimentalists to have the dLight1.3b sensor characterized alongside the six dopamine sensors described in the original manuscript version. The dLight1.3b sensor properties are described on page 5 lines 4-7 and integrated into Figure 1+2, Table 1+2, and Supplementary Figure 1. As described below in our response to reviewer 2, we have also added data from new experiments describing the sensitivity of all the sensors for noradrenalin (NA). This information is particularly important for experimentalists using experimental set-ups where both DA and NA are present. Finally, we have included new data on DAT-mediated dopamine uptake, acquired at a higher sampling rate, which further supports the use of sniffer cells in assays assessing DAT function.

“In Figure 3B, the F/F₀ response of the sniffer cells after 90 mM KCl is only 1.1. This is very strong stimulation and I am wondering the reasons.”

Thank you for this relevant comment. The F/F0 response in the neuronal cultures is highly dependent on the density, purity and maturity of the dopamine neuronal culture. While the recordings of dopamine release following electrical stimulations are made from dense rat dopamine cultures, the KCl recordings are made on scarcely seeded mouse dopaminergic neurons expressing td-tomato, which allowed us to positively identify individual neurons. In revisiting the manuscript, we realized that we have not made this point clear enough and we changed the text on page 13 lines 14-28 and adjusted Figure 4 accordingly.

“Throughout the manuscript, a space is required between numbers and units. For example, page 4: 10uM need to be 10 uM, page 6: figure legend, page 7: 4.3min⁻¹ to be 4.3 min⁻¹, etc, Page 4: head-to head, page 16: Hz to be Hz.”

We apologize for these mistakes. The manuscript has been revised to correct this.

Reviewer 2

“I enjoyed reading this manuscript. Overall, the work is, for the most part, sound, and experiments are well-designed. I appreciate the evaluation of DA sensors under the same experimental formats, providing much-needed information to experimentalists. I have some minor suggestions for clarifying some concerns in the manuscript. “

We thank the reviewer for the positive evaluation of our work. We have addressed the suggestions and concerns (see below).

1. All reported DA sensors had been shown to have cross-reactivity to noradrenaline (NA). In particular, the series of GRAB_DA, which are based on DRD2, exhibit only a little difference in sensitivity between DA and NA. Therefore, the data obtained in Figure 3 must also take into account not only DA but NA release from the striatum. If possible, it would be desirable to obtain a dose-response curve of six DA sensors for NA under the same conditions. If not, differences in sensitivity to NA should be mentioned in the revised manuscript.

We appreciate this very relevant comment. Indeed, the selectivity of each sensor for DA over NA is highly relevant for both in vivo and ex vivo experimentalists. In the revised manuscript, we have included data from new experiments deriving dose-response curves of all seven DA sensors for NA to determine the molecular selectivity of the sensors for this neurotransmitter. The results are described in the results section on page 5 line 28 to page 6 line 6 and are presented in Table 3 and Figure 2. Furthermore, we have added a discussion of this new data to the measurements of dopamine from striatal slices and tissue lysates (page 14 lines 16-20).

2. The authors should describe whether the Sniffer cells used in this study were single-cell cloned or a bulk population.

We utilized a Flp-In 293 T-REx cell line to develop the sniffer cells, meaning it is equivalent to a single-cell cloned population. As per the manual of the cell line, co-transfection of

the Flp-In™ Cell Lines with a Flp-In™ expression vector (in this case the DA sensors) and the Flp recombinase vector, pOG44, results in targeted integration of the expression vector to the same locus in every cell, ensuring homogeneous levels of gene expression. We have now changed the sentence in the results section page 4 line 23-24 to the following to make this important point clearer: “Of note, as the Flp-In system was utilized for the generation of the sniffer cell lines, all sensors were inserted at the same specific genomic location ensuring homogenous levels of gene expression.”

3. In Figure 2, it is intriguing findings that the response of the DA reporters is reduced in Sniffer cells expressing DAT and that DA is released through DAT. However, it seems unlikely that only tens of thousands of cells could rapidly uptake such a large amount of DA (probably 100 uL/well of 10 uM DA solution) to reduce the concentration of DA (Figure 2A). The authors should verify whether or not there is an artifact. For example, does the %Dopamine value just after DA stimulation approach 100% in DAT-expressing cells, if data are acquired at a higher sampling rate? If the conditioned medium 15 minutes after DA stimulation in DAT-expressing cells is collected and added to Sniffer cells that do not express DAT, will their response be about 30% of the %Dopamine value? If DAT-expressing and non-DAT-expressing cells are co-cultured and imaged under a microscope, is it possible to obtain the same level of response for DA sensors?

The reviewer is correct that the cells do not take up such a large amount of DA (200ul/well of 1uM DA solution). Thus, when we transfer conditioned media from hDAT expressing cells and empty (non-hDAT expressing) cells after 15 minutes of DA stimulation to a new plate of sniffer cells that do not express hDAT, we see no difference in the fluorescent response, confirming that it is not the global dopamine concentration that is reduced by the hDAT-driven dopamine uptake. Rather we believe the presence of DAT produces a local dopamine depletion at the cell surface where the sensors detect DA as well, thus reducing the amount of DA detected in an hDAT-dependent manner. Indeed, this is an important point as ligand depletion would challenge kinetic studies on DAT. We have now changed the text on page 10 line 7-9 to: “We rationalized that hDAT-mediated DA uptake would produce a local decrease in extracellular DA concentration near the cell surface and thereby reduce activation of nearby GRAB_{DA1M}.”. Moreover, as suggested by the reviewer, we have now included data that has been acquired at a higher sampling rate which indicates that just after DA stimulation, hDAT-expressing cells do indeed follow the same fluorescence increase as non-hDAT-expressing cells for a few seconds but never reaches the same level as control cells, consistent with hDAT-mediated local dopamine depletion. These data are presented in Supplementary Figure 4. Importantly, we are confident that the depletion of DA at the cell surface is hDAT-mediated, as the response can be inhibited by several hDAT inhibitors in a dose-dependent manner (Figure 3B), while the inhibitors do not affect the sensor in the absence of hDAT expression. If hDAT expression somehow affected the fluorescent response of the sensor in an unspecific, i.e. hDAT-independent, manner, this effect should not be dose-dependently sensitive to hDAT inhibitors.

4. Figure legend of Fig 2A says 1 uM DA, but the Y-axis of Fig 2A says 10 uM, 5 min. Which is correct and what is 5 min?

We apologize for this unclarity. The cells were stimulated with 1 μ M DA but normalized to a 5 min stimulation with 10 μ M DA. 10 μ M DA induces the maximal fluorescent change (as can be seen in Figure 1+2), and was utilized to ensure that the data between hDAT- and empty vector-expressing cells were comparable (as hDAT expression could alter the sensor expression and thus the change in fluorescence detected). We have now clarified this in the text (page 10 line 13-16) and Figure 3's legend by adding the following sentence: "To ensure that the measurements between hDAT- and empty vector-transfected cells were comparable (as the expression of hDAT could affect the sensor expression) the data were normalized to a 5 min stimulation with a saturating concentration of DA (10 μ M)."

Collectively, we believe that the corrections have strengthened the manuscript, and we would like to again thank the editor and reviewers for taking the time to handle and review the manuscript, which we hope will now be found suitable for publication.

Thank you for your kind consideration.

Sincerely,

Freja Herborg, PhD
Assistant Professor
Department of Neuroscience, University of Copenhagen, Denmark
Email: frejahh@sund.ku.dk

REVIEWERS' COMMENTS:

Reviewer #1 (Remarks to the Author):

I have checked the response to the reviews and the revised manuscript accordingly. I think the manuscript has been improved during this process.

Now I recommend this manuscript for publication in the journal.

Reviewer #2 (Remarks to the Author):

Overall, the authors have done a mostly adequate job of responding to the reviewers' comments.